# Effect of Sulphur-Containing Tailings Content and Curing Temperature on the Properties of M32.5 Cement Mortar

**DOI:** 10.3390/ma14195751

**Published:** 2021-10-01

**Authors:** Qian Chen, Haiming Chen, Pengju Wang, Xiang Chen, Jie Chen

**Affiliations:** 1School of Civil Engineering and Architecture, Anhui University of Science and Technology, Huainan 232001, China; 2019200230@aust.edu.cn (Q.C.); 2018200213@aust.edu.cn (P.W.); 2020200312@aust.edu.cn (X.C.); 2021200343@aust.edu.cn (J.C.); 2Engineering Research Center of Underground Mine Construction, Ministry of Education, Anhui University of Science and Technology, Huainan 232001, China

**Keywords:** curing temperature, sulphur-containing tailings, microstructure, sulfate, pore size distribution

## Abstract

The effect of the dosage of sulphur-containing tailings (STs) and curing temperature on the properties of M32.5 cement mortar was studied in this work. An experimental study was conducted to evaluate the effects of STs with different substitution ratios (0, 10%, 20%, 30%, 40%) on the compressive strength experiment, fluidity, expansion ratio, and pore structure of M32.5 cement mortar. The results showed that the addition of STs reduced the fluidity of mortar, and the fluidity decreased with the increase of the STs dosage. The compressive strength of mortars increased at a lower substitution rate (0~20%) but decreased at a higher substitution rate (>20%). Ettringite peaks and new sulfate peaks were found by X-ray diffraction (XRD) analysis. Scanning electron microscope (SEM) observation of the microstructure showed that a large number of hydrated products, such as ettringite, formed and filled in the interstitial space, which was conducive to the development of strength. The optimal STs replacement ratio of river sand was 10%. Then, the performance of mortar at curing temperatures of 23 ± 1, 40, 60, and 80 °C was further investigated under the optimal STs replacement ratio. Under high-temperature curing conditions, the early strength of M32.5 cement mortar with STs increased greatly, but the late strength decreased gradually with the increase in curing temperature. The early strength development of the mortar mainly depended on the high speed of hydration reaction, and the late strength variation was mainly affected by hydration products and the pore size distribution. After comprehensive consideration, the optimal curing temperature of M32.5 cement mortar with STs was 40 °C.

## 1. Introduction

Resource utilization of tailings is one of the key directions for green development of mines [1,2,3]. In recent years, domestic and foreign scholars have done a lot of work on the resource utilization of iron tailings and copper tailings, which promotes the utilization technology development and application of relevant tailings [4,5,6,7]. However, metal mines are accompanied by sulfide. The sulfide in the form of pyrite is prone to oxidation to produce sulfate, which has a significant impact on cement hydration. In general, in cement-based systems, excessive sulfate can react with hydration products to form ettringite, reducing calcium hydroxide (CH), and causing problems, such as the structure deterioration of calcium silicate hydrate (C-S-H) [8,9,10,11]. Therefore, how to avoid or reduce the impact of sulfate on cement hydration has become a key problem for the resource utilization of STs. For this reason, domestic and foreign scholars have proposed a utilization method of STs to prepare filling materials for mine. For example, Liu et al. [12] pointed out that tailings backfill materials were prepared by solid waste in mines to fill mined-out voids, which reduced the volume of the tailings dam, importantly protecting the environment and building environmentally friendly mines. Wang et al. [13] used agricultural waste rice straw ash (RSA) to partially replace cement with a view to reducing the filling cost and improving the strength performance of cemented tailings backfill. Jiang et al. [14] found that the tailings backfill with sulfur content less than 18% had strength greater than 1.2 MPa after 28 days and can be directly used as filling aggregate. The general idea of the method is to use fly ash, slag, calcium carbide slag, etc. to form composite excitation materials to cement STs to produce filling materials for the mine, and to realize in situ resource utilization of STs [15,16]. However, it has been found that this method has a relatively limited consumption of STs. So, it is particularly important to develop effective large-scale utilization technologies that can consume large amounts of STs to solve the problems of STs.

In the sulfate aggressive environment, ordinary Portland cement (OPC) is often affected by sulfate erosion, which leads to material deterioration [17,18]. OPC contains a lot of clinkers, such as tricalcium aluminate (C_3_A) and tetra-calcium aluminoferrite (C_4_AF), which react with sulfate to form dilatant hydration products, e.g., ettringite, that have adverse effects on the stability and strength of the mortar. M32.5 cement is a kind of composite cement that is different from OPC and is a kind of low-clinker green cement prepared by adding a large amount of fly ash, slag ash, and other composite materials. Therefore, the use of M32.5 cement with low clinker and cement STs may effectively avoid the incompatibility effect of sulfate produced by STs on cement-based materials, and a small amount of ettringite produced by cement hydration can fill the micro-pores of the mortar and is conducive to the development of strength.

Using M32.5 cement to cement STs may have the following benefits: (1) Make full use of the characteristics of low clinker and high auxiliary materials of this kind of cement to reduce the adverse effects of the conversion of pyrite and other sulfides in STs on cement hydration. (2) The sulfate formed by the conversion of STs can assist in stimulating the auxiliary materials in M32.5 cement, further enhancing the degree of cement hydration and contributing to higher strength. (3) Using M32.5 cement to cement STs helps to prepare green materials, which can be used for the water-stable layer and plastering mortar. Compared with OPC, M32.5 cement with STs has better cementing stability, which will not cause deterioration of mortar within a certain amount of STs.

However, the acidification reaction of pyrite in STs is slow at room temperature [19,20]. The strength improvement of the mortar with STs is not obvious. Pyrite can be converted into sulfate and iron trioxide (Fe_2_O_3_), in which sulfate ions play an auxiliary excitation role in the early stage, which can generate ettringite to improve the compressive strength of the mortar in the early stage [21]. Temperature can indirectly affect the strength of the cemented body by affecting the rate of hydration reaction. However, hydration products, such as ettringite, will decompose at around 60 °C [22], destroying the original dense microstructure, which leads to poor mechanical properties of the cemented body [23,24,25]. However, there has been no research on the influence of the curing temperature on the performance of M32.5 cement in China and abroad. Therefore, the selection of an appropriate curing temperature will have a favorable effect on the mortar with STs, which has a certain guiding significance for assisting in the evaluation of the acidification reaction of pyrite in STs. Nuclear magnetic resonance (NMR), SEM, and XRD technology were carried on to illustrate the mechanism of the effect of STs content and curing temperature on the properties of M32.5 cement mortar.

## 2. Materials and Methods

### 2.1. Raw Materials

The materials used in the experiments include M32.5 cement, STs, and sand. The apparent density of M32.5 cement is 3060 kg/m^3^. The used STs came from Dongguashan Copper Mine in Tongling City, Anhui province, China, and the apparent density is 3081 kg/m^3^. The most probable particle size of STs is 66.9~76.0 μm, as shown in Figure 1. The SEM image of STs is shown in Figure 2. Figure 3 shows that STs contains a variety of minerals, including pyrite, quartz, hematite, and sulfate, among which quartz and sulfate are the main mineral forms. Sand used in this work was collected from Huaihe river, China, whose fineness module and apparent density are 2.6 and 2650 kg/m^3^, respectively. The chemical composition of the raw materials is shown in Table 1. Common tap water was used as the mixing water.

### 2.2. Mix Proportions

To investigate the effect of STs content on the performance of M32.5 cement mortars, 5 mixtures were considered by adding STs into mortars replacing 0, 10%, 20%, 30%, and 40% sand by weight, which can be further found in Table 2. Con denotes the control group, the letter M denotes M32.5 cement, the letter S denotes STs, and the number denotes the percentage of STs replacing river sand. The water-binder ratio (w/b) is 0.45.

The effect of the curing temperature on the properties of M32.5 cement mortars mixed with STs was further investigated. In the experiment, the water-binder ratio is 0.45. The sand ratio of the control group (CRT) is 50%, and the optimal STs replacement ratio of river sand is 10% in the M32.5 cement mortars with STs according to the research result of STs content on the performance of M32.5 cement mortar. Therefore, 10% STs are used to replace river sand. As shown in Table 3, four different curing temperatures were used, namely, (23 ± 1), 40, 60, and 80 °C. CRT denotes M32.5 cement mortars without STs cured at room temperature, RT denotes M32.5 cement mortars with STs cured at room temperature, the letter T denotes the temperature, and the number denotes the temperature value.

### 2.3. Test Methods

The experiments were systematically designed to explore the effect of the dosage of STs and curing temperature on the properties of M32.5 cement mortar. According to ASTM C1437-20 [26], the table flow test was used to determine the fluidity of mortar specimens. The mortar was mixed into the table flow test die and tamped from the edge to the center by the tamp bar. After the tamping was complete, the die sleeve was removed, and the tangent cone was slightly lifted round the die in the vertical direction. The table was immediately started, once every second, and 25 jumps were completed in 25 ± 1 s. After the jumping was completed, the caliper was used to measure maximum diffusion diameter of the bottom surface of the mortar and its vertical diameter, then its mean value was calculated. The table flow value was evaluated as a percentage of increased average base diameter relative to the original base diameter of fresh mortar.

The compressive strength was measured after curing for 3, 7, and 28 days using 50-mm-cubical mortar specimens at 23 ± 2 °C and a humidity of more than 95% according to ASTM C109/C109M-20b [27]. Each batch contained triplicate samples and average results were reported.

The expansion of the mortar specimens was determined according to ASTM C806-18 [28]. The mortar was placed in molds of 50 by 50 by 250 mm, and then the specimens were placed in the curing room. The specimens were removed from the molds at an age of 6 ± 1/4 h and the initial comparator reading *L_i_* was recorded. The specimens were cured in lime-saturated water at 23.0 ± 1.7 ℃ until they reached an age of 3, 7, 28, and 56 days, and the comparator reading of specimens at x age *L_x_* was recorded. The expansion of the mortar specimens was calculated according to Formula (1):(1)$$E_{x}=\frac{L_{x}-L_{i}}{10\times 250}\times 100$$
where:

*E_x_* = the expansion at x age, %;*L_x_* = the comparator reading of specimen at x age, mm; and*L_i_* = the initial comparator reading of restraining rod, mm.

The microstructural characteristics of the M32.5 cement mortar incorporating STs were evaluated by means of SEM, XRD, and nuclear magnetic resonance (NMR). A Flex1000 scanning electron microscope with an acceleration voltage of 15 kV was used for the SEM test. The samples were immersed in absolute ethyl alcohol for one week. Then, the samples were taken out and put in a vacuum pump for vacuum treatment for 24 h. Before the test, a thin layer of gold was sputtered on the sample with an MSP-2S magnetron ion diffractometer for 90 s to obtain the conductivity. The XRD analysis was conducted using a BD68000162-01 X-ray diffractometer by employing CuKα radiation at 40 kV and 50 mA, a scan speed of 5°/min, and a scan range of 5 to 60°. Samples used for XRD analysis were pulverized paste specimens at 28 days, which were put in a vacuum pump for 48 h to prevent moisture retention. The filter needed to be cleaned with anhydrous ethanol before use, and then the powder samples were pressed flat and placed in the middle of the filter so that the CuKα rays could evenly scan the sample. The MesoMR23-060V-1 Newman NMR spectrum was used for the pore structure test. Specimens were prepared using a cylindrical mold with a diameter of 50 mm and a height of 50 mm, and three specimens were made in each group. The magnetic field intensity of the spectrometer was 0.5 ± 0.05 T, and the resonance frequency of the corresponding proton was 21.3 Hz. The temperature was maintained at 32 °C throughout the experiment.

## 3. Results and Discussion

### 3.1. Effect of STs Content on The Properties of M32.5 Cement Mortar

#### 3.1.1. Fluidity

The fluidity of the fresh mortars with STs decreased with the increase in the amount of STs (Figure 4). Compared with the control group (without STs), the fluidity reductions of the mortars with STs were 8.53%, 19.1%, 32.1%, and 42.6%, respectively. The decrease in the fluidity of the mortars with STs can be attributed to the fact that STs have greater water absorption than river sand. STs absorb more water than river sand, resulting in a decrease in the free water content and lower fluidity of the fresh mortars. In addition, the SEM picture of STs, as shown in Figure 2, suggested that the ST particles were irregular prismatic, with different sizes, loose surface structure, and no densification, resulting in large friction between ST particles, which significantly reduced the fluidity of fresh mortars. Therefore, it is necessary to limit the substitution rate of STs to river sand; otherwise, an excess substitution rate will cause poor fluidity of the mortars.

#### 3.1.2. Compressive Strength

Figure 5 presents the compressive strength of M32.5 cement mortars with different amounts of STs. When the curing time increased from 3 to 28 days, the overall compressive strength of the five groups of mortars tended to increase. After adding STs, the compressive strength of mortars increased at a lower substitution rate (0~10%) but decreased at a higher substitution rate (>10%). The positive effect of STs on the compressive strength of mortars can be explained from two aspects: (1) STs replace river sand, which is equivalent to reducing the sand rate and increasing the amount of cementitious materials. In the case of constant water consumption, the water-binder ratio decreases accordingly, and the mortar structure is denser, thereby improving the compressive strength of mortar. (2) Due to the release and migration of sulfur in STs, sulfate ions generated by the reaction of pyrite assist in activating the cement auxiliary material, promote the hydration reaction, and thus improve the compressive strength of mortars [29,30].

However, the addition of STs also has a negative effect on the compressive strength of mortars. Pyrite in STs is prone to oxidation to produce sulfate, which will react with CH and C_3_A to form dilatant hydration products, e.g., ettringite, that have adverse effects on the strength of mortar. As the content of STs increases, the dense structure of the mortar is destroyed, leading to deterioration of the bonding force between the aggregates, and a reduction of coarse-grained aggregates weakens the framework of river sand and reduces the compressive strength of mortar. The increase in the ST content increases the specific surface area of the particles per unit volume of the mortar. Since the activity level of STs is much lower than that of cement, more hydration products are needed to provide cohesive force. When the STs content exceeded 10%, the compressive strength decreased gradually.

#### 3.1.3. Phase Analysis

Figure 6 shows the XRD patterns of hydration products of mortars with STs cured for 7 and 28 days. Diffraction peaks corresponding to ettringite (Aft), calcium hydroxide (CH), pyrite (FeS_2_), quartz (SiO_2_), calcite, and gypsum can be observed. Aft peaks at diffraction angles of 9.091° and 15.784° were detected in trace amounts in mortars with STs, but it was not observed in mortars without STs, which means the addition of STs in mortar produced ettringite. Compared with the Con group, gypsum was also detected at the diffraction angle of 33.346° in the MS groups, and secondary gypsum can further promote the formation of ettringite.

New sulfate mineral (Sepierite) peaks were observed in the MS groups, which indicated that the sulfur in STs had been transformed to form sulfate ions, and then produced sulfate mineral. The peak value of sulfate increased with the increase of the dosage of STs, indicating that the new sulfate mineral formed after more STs were incorporated [31]. The sulfide in STs is mainly ferrous sulfide (FeS_2_). It can be seen from Figure 6b that the peak value of FeS_2_ gradually became smaller at 28 days, indicating that the sulfide had reacted, that is, it was converted to form sulfate ions. Correspondingly, the peak value of sulfate increased with the curing time.

#### 3.1.4. Microstructure Analysis

Figure 7 shows the SEM pictures of the mortars at 28 days. As shown in Figure 7a, only a small amount of ettringite was observed in the con group. After adding STs, it can be found that flocculent C-S-H gels distributed around the pores, and large amounts of ettringite were generated. These hydration products filled the cracks, making the structure of mortar more compact (Figure 7b–d). A large number of needle-like clusters of ettringite were gathered together to fill the gaps between the aggregate particles and facilitated the development of compressive strength.

The incorporation of STs increased the amount of ettringite generated, indicating that the sulfur in STs promoted the formation of sulfate ions, and the formation of sulfate ions assisted in stimulating the hydration reaction of cement mortar, resulting in an increase in the amount of sulfate hydration products, such as ettringite. An appropriate amount of ettringite can refine the pores of compact mortar, but excessive ettringite will swell the microstructure of mortars, causing more cracks in mortars, leading to deterioration of the microstructure, and seriously affecting the strength development of mortars. It can be seen from Figure 7c–d that there were some cracks near the hydration product (flocculent gel and ettringite), indicating that the mortars had deteriorated after adding too large a dosage of STs, which is unfavorable to the strength development of mortars.

#### 3.1.5. Expansion

Figure 8 suggests that the addition of STs can cause mortars to expand. The sulfur in STs was released and converted to sulfate ions, which promoted the formation of expansive hydration products, such as ettringite [32,33]. With the increase in the replacement rate of STs, the expansion ratio of mortars first increased and then decreased. This is because sulfate ions have the effect of promoting cement hydration, consuming more water, and the consumption of water can form pores. To maintain the force balance between the pores and the solution, the capillary tension in the liquid can only be balanced by the compressive stress of the particles around the pores, which will cause the pores to be compressed, and make the expansion of mortars decrease. Besides, drying shrinkage of mortars also led to the expansion of mortars decreasing.

It can also be seen from Figure 8 that the expansion ratio of mortars first increased and then decreased with the increase of the curing time. At the early curing time, the formation of expansive hydration products, such as ettringite, caused the increase of the expansion ratio. At the later curing time, compared with the river sand, STs have a smaller particle size, so the STs have larger specific surface area, and the mortar has stronger water absorption, resulting in a higher water content, larger drying shrinkage, and a decrease of the expansion ratio. In general, the expansion of the mortar belongs to micro expansion, which will not affect the stability of the mortar.

### 3.2. Effect of Curing Temperature on The Properties of M32.5 Cement Mortar with STs

#### 3.2.1. Compressive Strength

As shown in Figure 9 and Table 4, the compressive strength of RT was bigger than that of CRT under room temperature ((23 ± 1) °C). Compared with CRT, the compressive strength of RT increased by 4.24%, 21.61%, and 11.62% at 3, 7, and 28 days, respectively, indicating that STs plays a positive role in M32.5 cement mortar. However, the strength improvement of the mortar with STs is not obvious under room temperature. This is because the acidification reaction of pyrite in STs is slow at room temperature.

As the curing temperature increased, the strength improvement of the mortar with STs became more obvious (Table 4). For example, compared with RT, the compressive strength of T40, T60, and T80 at 3 days were 32.73, 36.49, and 28.20 MPa, and increased by 99.93%, 122.86%, and 72.23%, respectively. Figure 9 suggests that with the increase in the curing temperature, the compressive strength of the mortars with STs increased at first and then decreased. On the one hand, high-temperature curing can accelerate the process of cement hydration, forming more hydration products in a relatively short time, so that the compressive strength can be greatly improved in a short time. On the other hand, high temperatures can also accelerate the oxidation of sulfide to form sulfate, which assists in stimulating the activity of the slurry and further enhances the strength [34]. However, a too high curing temperature reduced the compressive strength of mortar, for example, at 28 days, the compressive strength of T80 decreased by 12.92% compared with RT. At the late curing age, the high temperature affected the internal structure of mortar, causing the hydration products to be destroyed, and the original compact pore structure became loose, resulting in deterioration of the slurry, and a decrease of the compressive strength of the mortars.

#### 3.2.2. Microstructure Analysis

To explore the mechanism of mortar strength changes under different curing temperature, the microstructure of each specimen cured for 28 days was analyzed (Figure 10a–d). No large amount of hydration products was observed on the microstructure surface of RT, which was related to the fact that M32.5 cement contained less clinker (C_3_A, C_4_AF) and the acidification reaction of pyrite in STs was slow at room temperature. Under standard curing conditions, there were fewer hydration products, which cannot effectively fill in the pores, and the cracks were obvious.

However, at the high curing temperature, a large amount of ettringite and flocculent C-S-H gel were detected on the surface of the specimens (Figure 10b–d). As shown in Figure 10b, a large number of acicular ettringite crystals were scattered on the surface of the microstructure, and hydration products, such as gel, were observed, which means the hydration was sufficient. The hydration products were filled with each other, and the microstructure of T40 was complete and dense, so that the compressive strength of T40 was improved.

When the curing temperature was 60 °C, at the same time as the formation of ettringite, the high temperature promoted its decomposition and the microstructure surface was complete, but cracks were detected (Figure 10c), which caused the compressive strength of T60 to be lower than that of T40.

With the increase in the curing temperature, the hydration rate of the mortar was accelerated, and the degree of hydration was enhanced. However, under high curing temperatures, the specimen would deteriorate. Figure 10d shows that a large number of acicular ettringite and C-S-H gel was distributed on the surface of the microstructure, which were the products formed by cement hydration. Nevertheless, at the same time, large cracks appeared on the surface of the microstructure, resulting in a loose structure, lack of compactness, and deterioration of the specimens, so that the compressive strength decreased [35,36,37].

#### 3.2.3. NMR Analysis

Under different curing temperatures, the hydration rate of mortar increased with the increase in temperature, but the compressive strength did not show a similar trend. Therefore, the reason for the strength change can be analyzed from the characteristics of porosity and pore size distributions because they are considered as major factors influencing the strength of mortars. The curve of the specimen at 28 days (Figure 11) revealed that the maximum continuous pore diameters of all four curing conditions were nearly the same, approximately 60 nm, which belongs to harmful pores (20–100 nm) [38,39,40]. The higher the peak value, the worse the pore distribution. With the increase in the curing temperature, the peak value of harmful pores became larger. The peak value of harmful pores of T40 was close to that of RT, indicating that the curing temperature of 40 °C had little effect on the pore distribution but increased the hydration rate, so the compressive strength increased. However, when the curing temperatures were higher, the peak values of harmful pores corresponding to T60 and T80 were larger, which were 3 to 4 times that of RT, and the pores distribution was poorer, resulting in a decrease of the compressive strength.

An appropriate curing temperature increase can promote the hydration reaction and generate expansive hydration products to fill the pores, so that the number of gel pores increases (Figure 12a). When the curing temperature was 80 °C, the hydration reaction rate was the fastest and the degree of hydration was the highest. Although the generated hydration products can fill part of the pores, the negative effect of the high curing temperature was greater. The high-temperature expansion of the specimens caused cracking of the microstructure surface and an increase of harmful pores [41]. High curing temperature can promote the rate of hydration reaction but cannot increase the hydration products, which cannot fill a large number of cracks generated by the high curing temperature, resulting in a decrease of the strength of the mortars.

The increase of the curing temperature obviously increased the volume proportion of harmful pores (Figure 12b), and slightly increased the ratio of gel pores. The hydration products produced by the temperature increase can improve the distribution of larger pores to some extent. Considering this comprehensively, when the curing temperature was 40 °C, fewer harmful pores were generated, and the volume of gel pores increased, resulting in less damage to the microstructure of T40, which explains why the compressive strength of T40 was the highest.

#### 3.2.4. XRD Analysis

Figure 13 shows the XRD patterns of M32.5 cement mortar with STs cured for 3, 7, and 28 days at room temperature, 40, 60, and 80 °C. Ettringite and sulfate peaks were observed in hydration products of mortar cured for 3 days (Figure 13a). At room temperature and 40 °C, the hydration reaction was normal, and ettringite peaks were detected obviously at the diffraction angles of 9.091° and 15.784°. However, with the increase in the curing temperature, the ettringite peaks almost disappeared in the XRD patterns of T60 and T80, and instead a new sulfate peak was observed near 10°. Under the condition of high curing temperatures (60–80 °C), the ettringite will decompose, and a new chemical reaction will occur to form the monosulfur hydrated calcium sulfoaluminate (Afm) [42]. At the same time, Fe_2_O_3_ was found to be generated under the high curing temperature, which is because FeS_2_ in STs oxidated and was converted into Fe_2_O_3_ and sulfate ions that corresponded to the sulfate peak.

With the increase in the curing temperature at different curing ages, the peak value of Ca(OH)_2_ in the XRD patterns decreased significantly (Figure 13). The peak value of Ca(OH)_2_ of T60 was even higher than T40 cured for 3 days, indicating that in the early stage with the increase in the curing temperature, the appropriate high curing temperature is conducive to the hydration reaction, and more Ca(OH)_2_ is generated. When the curing temperature was 80 °C, the hydration reaction was rapid and sufficient, secondary hydration reaction occurred, and part of Ca(OH)_2_ was consumed to generate hydration products, such as ettringite. However, ettringite decomposed at high temperatures, so the peak of ettringite cannot be found in the XRD pattern at T60 and T80. The hydration products at 7 and 28 days showed little difference from those at 3 days. At 28 days, the decrease of Ca(OH)_2_ at 80 °C was the largest, and the peak value of C_3_S was the highest. In general, a high curing temperature can accelerate the hydration reaction and pyrite reaction and promote the secondary hydration reaction to generate a large number of expansive hydration products. As the curing age increased, the types of hydration products did not change.

## 4. Conclusions

This paper investigated the effect of the dosage of STs and curing temperature on the properties of M32.5 cement mortar. The following conclusions are drawn based on the experimental results:The compressive strength of mortars with STs increased at a lower substitution rate (0~20%) but decreased at a higher substitution rate (>20%) with the increase in STs content; adding STs reduced the fluidity of mortar, and the fluidity decreased with the increase of the ST dosage.The addition of STs induced the formation of sulfate ions, promoted the cement hydration reaction, and produced a large amount of swelling hydration products, such as ettringite. At a lower dosage (0~20%), it can fill the gap of big pores and increase the strength. However, at a higher dosage (>20%), expansion will occur, destroying the structure of slurry, and reducing the strength.The compressive strength of mortars with STs increased at first and then decreased with the increase of the curing temperature. High curing temperatures can accelerate the speed of the hydration reaction, but too high a temperature will cause the formation of more harmful pores and cracks.Under high curing temperatures, the early strength development of the mortar mainly depended on the speed of the hydration reaction, and the late strength change is mainly affected by hydration products and the pore size distribution.Proper dosage of STs will not cause deterioration of cement mortar in the M32.5 cement system, and these products can be used in the practical application of plaster mortar, road water stabilization, and other aspects. The main practical significance of STs partially replacing river sand is that it can consume a mass of accumulated waste tailings, save a lot of maintenance costs, and have economic and environmental benefits. In the work, the optimal STs replacement ratio of river sand was 10% and the optimal curing temperature was 40 °C.


## Figures and Tables

**Figure 1 materials-14-05751-f001:**
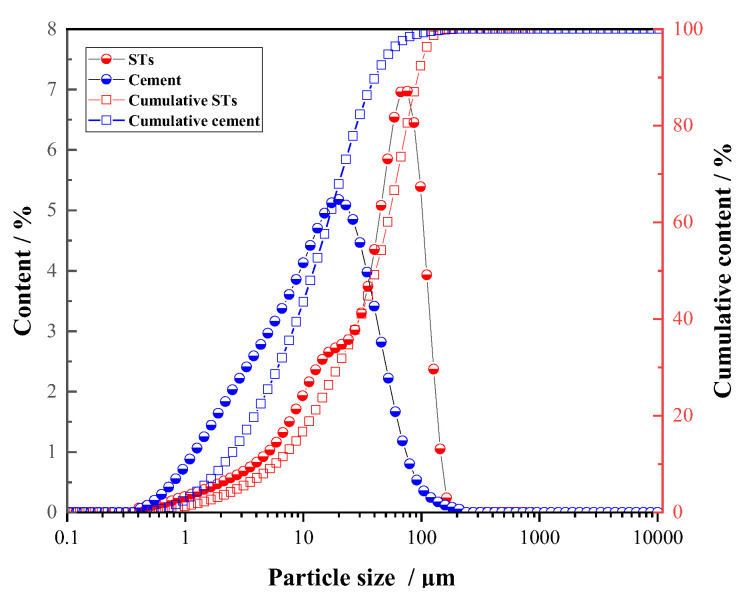
Particle size distribution curve of M32.5 cement and STs.

**Figure 2 materials-14-05751-f002:**
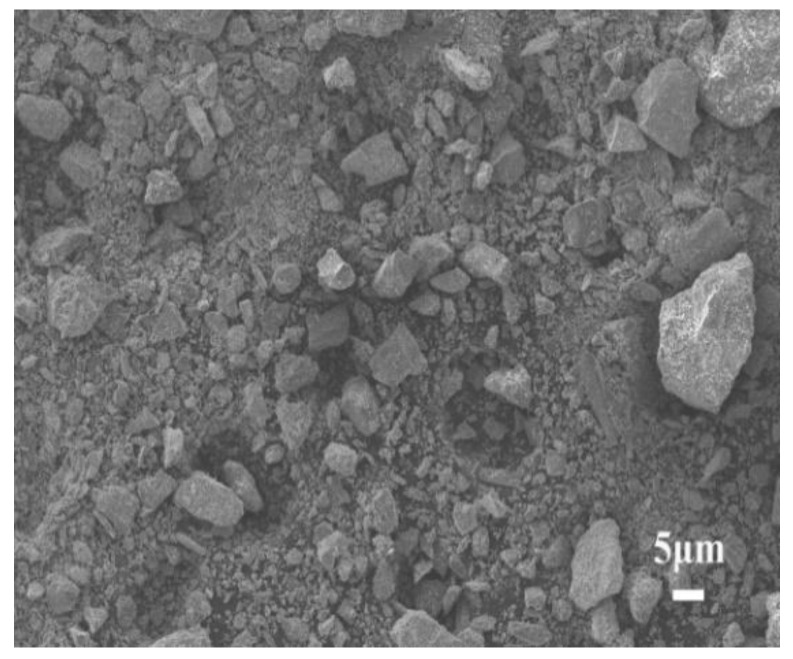
SEM image of STs.

**Figure 3 materials-14-05751-f003:**
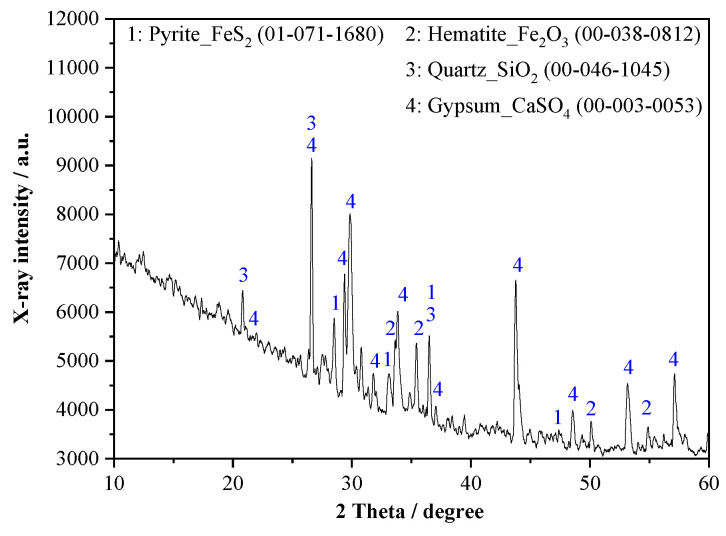
Chemical characteristics of STs obtained using XRD.

**Figure 4 materials-14-05751-f004:**
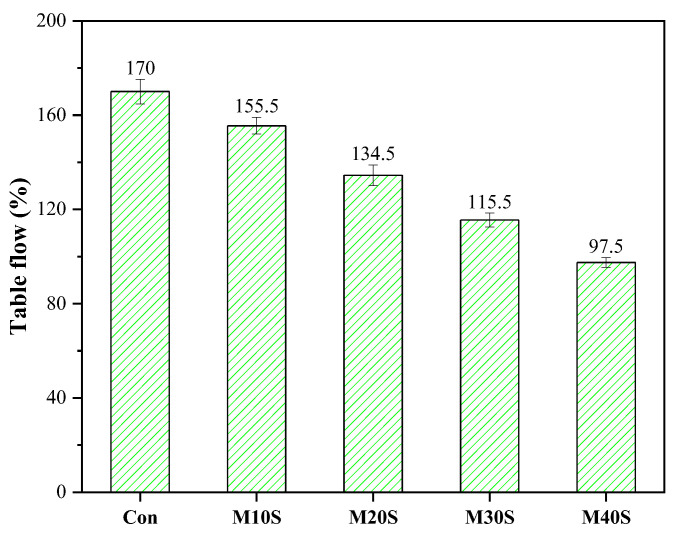
The table flow test results of the fresh mortars.

**Figure 5 materials-14-05751-f005:**
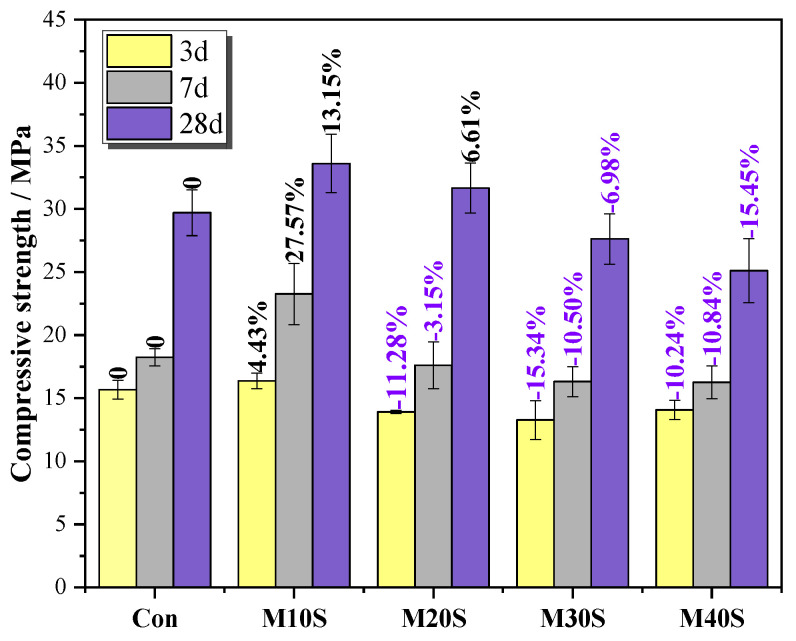
Compressive strength of mortars at 3, 7, and 28 days.

**Figure 6 materials-14-05751-f006:**
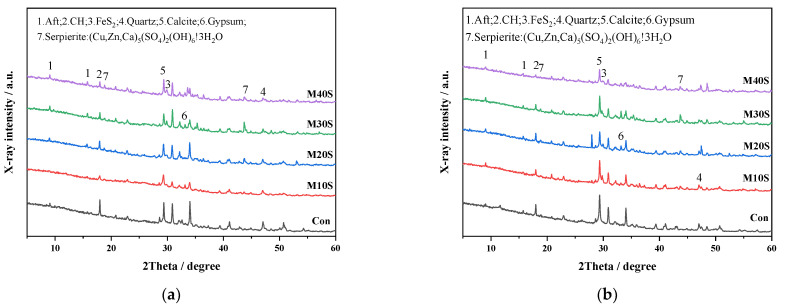
XRD patterns of hydration products of mortars with STs at different curing times: (**a**) 7 and (**b**) 28 days.

**Figure 7 materials-14-05751-f007:**
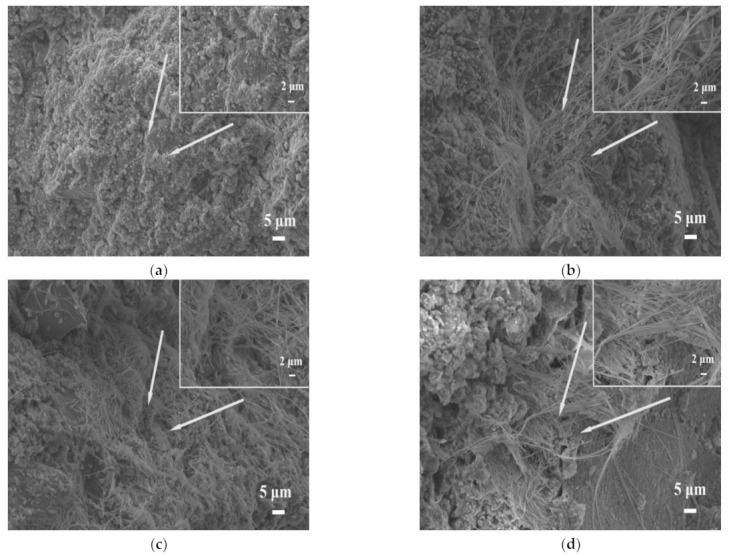
SEM images of mortars: (**a**) Con, (**b**) M10S, (**c**) M20S, (**d**) M30S.

**Figure 8 materials-14-05751-f008:**
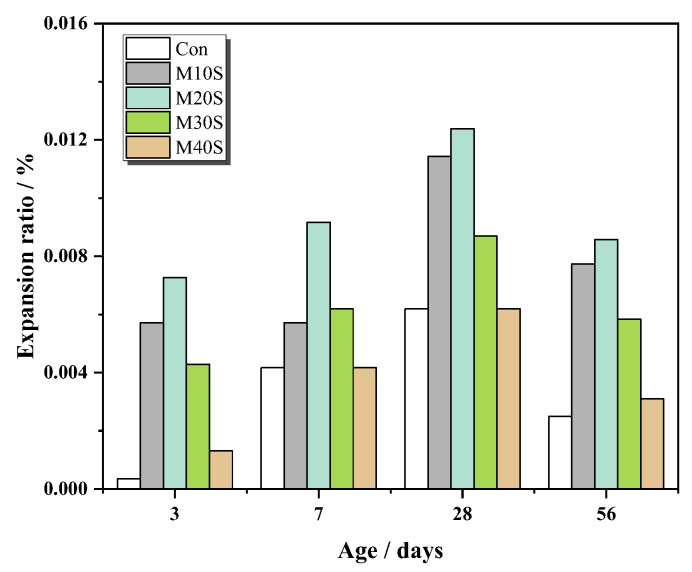
Expansion ratio of mortar specimens.

**Figure 9 materials-14-05751-f009:**
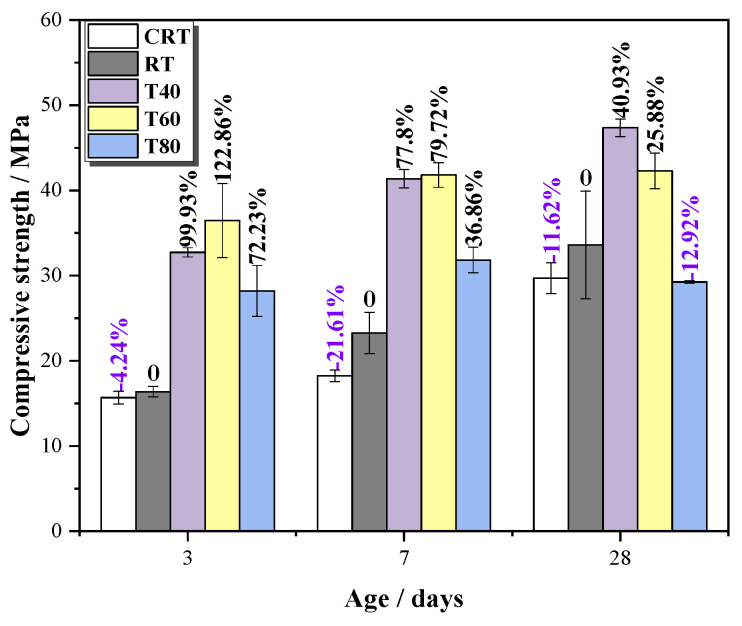
Compressive strength of mortars cured for 3, 7, and 28 days.

**Figure 10 materials-14-05751-f010:**
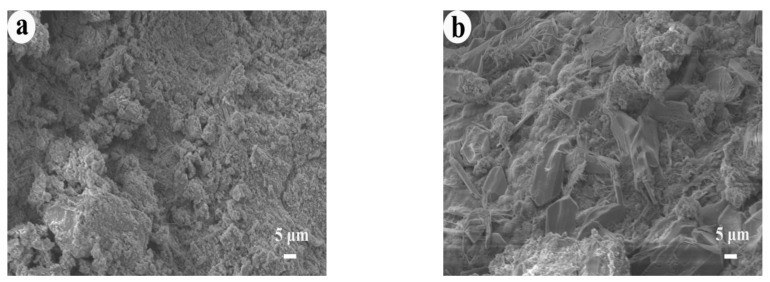
SEM images of mortars: (**a**) RT, (**b**) T40, (**c**) T60, (**d**) T80.

**Figure 11 materials-14-05751-f011:**
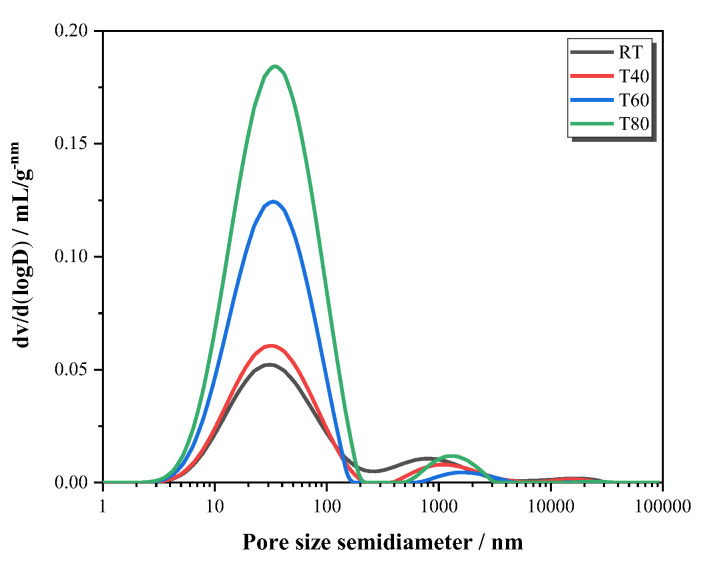
Pore volume curve of mortars at 28 days.

**Figure 12 materials-14-05751-f012:**
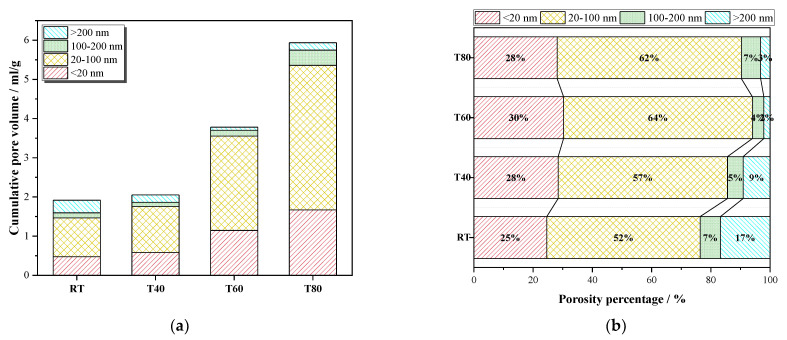
Pore distribution of mortars at 28 days: (**a**) pore volume distribution diagram, (**b**) pore distribution ratio diagram.

**Figure 13 materials-14-05751-f013:**
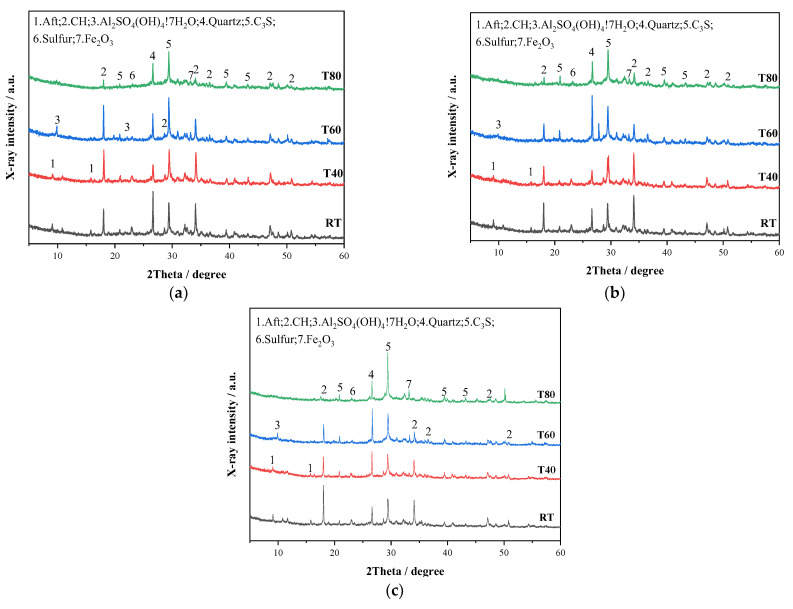
XRD patterns of mortars at different curing temperatures: (**a**) 3, (**b**) 7, (**c**) 28 days.

**Table 1 materials-14-05751-t001:** The main chemical composition of M32.5 cement and STs (wt./%).

Material	Al_2_O_3_	SO_3_	SiO_2_	Fe_2_O_3_	CaO	TiO_2_	MgO	Na_2_O	K_2_O	CO_2_	Other
M32.5 Cement	11.5	–	22.1	3.1	42.0	0.5	0.7	0.2	1.1	16.4	2.4
STs	5.2	21.1	22.1	30.5	10.3	0.2	5.0	0.6	0.8	–	4.2

**Table 2 materials-14-05751-t002:** The mix proportion of mortars for investigating the effect of STs content on the properties of mortars (g/100 g).

Number	M32.5 Cement	STs	Sand	Water
Con	34.5	0.0	50.0	15.5
M10S	34.5	5.0	45.0	15.5
M20S	34.5	10.0	40.0	15.5
M30S	34.5	15.0	35.0	15.5
M40S	34.5	20.0	30.0	15.5

**Table 3 materials-14-05751-t003:** The mix proportion of mortars for investigating the effect of curing temperature on the properties of mortars (g/100 g).

Number	M32.5 Cement	STs	Sand	Water	Curing Temperature/°C
CRT	34.5	0.0	50.0	15.5	23 ± 1
RT	34.5	5.0	45.0	15.5	23 ± 1
T40					40
T60					60
T80					80

**Table 4 materials-14-05751-t004:** The compressive strength of mortars cured for 3, 7, and 28 days (N/mm^2^).

	Number	CRT	RT	T40	T60	T80
Age						
3 d		15.68	16.37	32.73	36.49	28.20
7 d		18.24	23.26	41.36	41.81	31.84
28 d		29.70	33.60	47.35	42.30	29.26

## Data Availability

Data Sharing is not applicable.

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
