# Peer review of "Effect of Sulphur-Containing Tailings Content and Curing Temperature on the Properties of M32.5 Cement Mortar"

_materials, 2021, doi:10.3390/ma14195751_

Round 1
Reviewer 1 Report
Dear authors,
thank you for having the opportunity to review your paper. In general the focus of presented research and experiments provided were clear to me. I put several comments directly to your manuscript including also few questions. What was not fully clear to me if the part of experiments focusing on curing ws done only for the variant where 10% STs was included and if it is possible to explain why only this variant was checked and e.g. not verson with 20%. In conclusions I made a remark that it woudl be interesting to the reader to have also some opinion or your comment if it is useful to use STs in mortars and concretes, especially when you got all the findings with creation of entringitte and risks which are usually associated with it.

Author Response
Please see the attachment,attachment include the revised manuscript, Response to Reviewer Comments

Reviewer 2 Report
The paper presents the results of an experimental test campaign to assess to evaluate the effects of STs with different substitution ratios (0, 10%, 20%, 30%, 40%) on compressive strength, fluidity, expansion ratio and pore structure of M32.5 cement mortar. The topic is very interesting (experimental test results are always welcome) and could be suitable for the Journal, that has a very huge amount of paper published on the subject (two papers only, published in this journal, have been suggested for the references).
This referee thinks that the paper in its actual form is difficult to read and follow, so that the authors are invited to completely revise the English text with the help of a native English and to reformulate the paper after addressing all the issues underlined (the large use of the wh-word which must be checked). This referee has tried to be useful, but an extended revision of the text need a exceptional effort, outside the main task of this revision. Moreover, for future revisions line numbering would be appreciated.
Paragraph 1. “Introduction”
The presence of chemical formulas in the introduction section is perhaps to be avoided. Please, limit the chemical composition to a reference in parentheses and write the chemical name of the compound in the text.
Paragraph 2. “Materials and methods”
Paragraph 2.3.1, 2.3.2, 2.3.3, 2.3.4 and 2.3.5 should be rewritten without imperative forms. Moreover, please better explain the flow test (see for example https://doi.org/10.3390/ma12081322). It is determined in general as a percent increase in the diameter of a conically moulded mortar, but the details given by the authors seem different, therefore the picture in figure 4 is to be amended (...what is fluidity?).
Paragraph 3. “Results”
In general, the consideration about the test results should be based on data. A sentence like ”As curing temperature increased, the strength improvement of the mortar with STs became more obvious” seem a little obscure. Could the authors address the considerations to the numerical data? For example, observing Figure 10, the compressive strength increase with time is somewhat similar for RT and T40 mortars, lower than the CRT mortars, while a similar behaviour cannot be observed in the remaining two types. Could the authors discuss this point? A simple consideration like “with the increase in curing temperature, compressive strength of mortars with STs increased at first then decreased” is not sufficient. In general, in good conditions, a sensible increase of strength with time can be observed (see for example https://doi.org/10.3390/ma14030598), so that a decrease should be carefully taken into account, considering that the tests have been carried out at 28 days at the maximum. The life of a mortar is larger and the authors have certainly considered that environment conditions during the on site curing time are strongly variable, so that suitable indications should be given by the manufacturer. Could the authors address this issue? An appendix with the detailed numerical results for every tests could be useful for the readers.
Paragraph 4. “Conclusions”
The discussion re-arranged according the above suggestions. The authors should underline the novelty of their approach and address the above issues.
Minor corrections:
The authors are invited to revise all the text, as above suggested, nevertheless some minor corrections are reported in the following:
Page 1 “… However, metal mines are accompanied by sulfide, which causes the sulfide content to be excessive. Especially the sulfide ….”. ” Awkward. Please reformulate
Page 1 “…filling material for mine…” is not clear. Please clarify.
Page 2 “pavement water-stable layer” the position of the word pavement is perhaps not correct. Please check and reformulate the entire phrase
Page 2 “However, hydration products such as ettringite will decompose around 60°C, which affects the properties of the cemented body” is difficult to follow. Please reformulate
Page 2 “at home and abroad” please write “in China and abroad”
Page 2 “…whose apparent density are ……. The SEM image of STs as shown” The verb is not correct or absent. Please check
Page 4 “…In order to further study the effect of curing temperature on the properties of M32.5 cement mortars with STs.” The verb is lacking
Page 4 “…the STs replacement ratio of river sand is 10% in the M32.5 cement mortars without STs according to the research result of STs content on the performance of M32.5 cement mortar” The phrase is not clear. Please clarify
Table 3 Why the ratios are expressed by numbers with three decimal paces? Please clarify
Page 4 “…jumping table test” Is that the flow table test? Please clarify
Page 4 “…calculate it mean value” Perhaps it is “its”.
Page 4 “…after curing 24h” Please correct.
Page 6 “…It was seen from Figure 4” bad. Please amend
Page 6 “…the degree of reduction in compressive strength is significantly improved” Perhaps it is not a great improvement……could the authors find a better word? Is it increase?.
Page 6 “…and a large number of ettringite were generated. These hydration products were filled the cracks” Please correct.
Page 11 “…hydration was more sufficient” Obscure. Please clarify.
Author Response

(The authors gave the same response as above.)

Reviewer 3 Report
The paper is quite well structured and the experimental results are clearly presented and procedures are properly designed.
Some minor corrections are required.
A single main error is identified when authors write that ettringite is the product of reaction between sulphates and calcium silicates, while it comes from calcium aluminates and sulphates.
Can the authors give more details on M32.5 cement composition?
Table 1 contains EDS results with too many decimal figures, usually the precision of such measurement is around 1%.
In the description of raw materials very high density values are indicated, can the authors describe the density measurement method? The apparent density of a powder or a sand is usually quite lower, maybe it's skeletal or true density instead of apparent one?
More details are required in the description of XRD measurement, radiation used, filters etc.
The last sentence in SEM description should be corrected.
The equation used in expansion calculation is unclear: the denominator 10(250)?
In figure 4 the error bars are missing, if possible include them.
The readability of figures 6 and 14 can be improved by substituing the numbers indicating the identified mineral with letters. Moreover a clear indication of curing time is needed. Finally it's very important describing the identified sulphate as general sulphates indication is not enough in XRay diffractometry. Please indicate the identified sulphate mineral.
Some minor mistakes are present in the first paragraph of page 8 "hydration products were filled the cracks" "mortar paticles and, which was conductive"
Finally in the discussion of XRD analysis results of samples cured at different temperatures the reduction of Ca(OH)2 peaks is associated with idration reaction leading to more CSH, but it's unlikely the case as we're not in presence of pozzolanic materials, Ca(OH)2 is probably involved in formation of ettringite.
Author Response

(The authors gave the same response as above.)

Round 2
Reviewer 2 Report
This referee asks for a revision made by a native English speaker, since the reading of the paper has not significantly improved after the changes